# Transcriptome Analysis of *Fusarium* Root-Rot-Resistant and -Susceptible Alfalfa (*Medicago sativa* L.) Plants during Plant–Pathogen Interactions

**DOI:** 10.3390/genes13050788

**Published:** 2022-04-28

**Authors:** Wenyu Zhang, Zicheng Wang, Zhencuo Dan, Lixia Zhang, Ming Xu, Guofeng Yang, Maofeng Chai, Zhenyi Li, Hongli Xie, Lili Cong

**Affiliations:** Key Laboratory of National Forestry and Grassland Administration on Grassland Resources and Ecology in the Yellow River Delta, Qingdao Agricultural University, Qingdao 266109, China; zwy06102121@126.com (W.Z.); fzqwzc@163.com (Z.W.); dzczynjy@163.com (Z.D.); aazllx@163.com (L.Z.); xuming199836@163.com (M.X.); yanggf@qau.edu.cn (G.Y.); chimu2100@126.com (M.C.); lizhenyily@163.com (Z.L.); hlxie08@126.com (H.X.)

**Keywords:** alfalfa, *Fusarium* root rot, transcriptome analysis, plant–pathogen interactions, disease resistance gene

## Abstract

Alfalfa (*Medicago sativa* L.) is a perennial leguminous forage cultivated globally. *Fusarium* spp.-induced root rot is a chronic and devastating disease affecting alfalfa that occurs in most production fields. Studying the disease resistance regulatory network and investigating the key genes involved in plant–pathogen resistance can provide vital information for breeding alfalfa that are resistant to *Fusarium* spp. In this study, a resistant and susceptible clonal line of alfalfa was inoculated with *Fusarium proliferatum* L1 and sampled at 24 h, 48 h, 72 h, and 7 d post-inoculation for RNA-seq analysis. Among the differentially expressed genes (DEGs) detected between the two clonal lines at the four time points after inoculation, approximately 81.8% were detected at 24 h and 7 d after inoculation. Many DEGs in the two inoculated clonal lines participated in PAMP-triggered immunity (PTI) and effector-triggered immunity (ETI) mechanisms. In addition, transcription factor families such as bHLH, SBP, AP2, WRKY, and MYB were detected in response to infection. These results are an important supplement to the few existing studies on the resistance regulatory network of alfalfa against Fusarium root rot and will help to understand the evolution of host–pathogen interactions.

## 1. Introduction

Alfalfa (*Medicago sativa* L.), known as the “queen of forage”, has become one of the most important forage crops globally because of its high yield, high quality, satisfactory palatability, and wide adaptability. Alfalfa root rot caused by *Fusarium* spp. is a major root disease affecting alfalfa production. After infection, the root neck and main root xylem gradually decay and become hollow, deteriorating the ability of the roots to absorb nutrients and water, which causes the plants to gradually die [1].

*Fusarium* root rot is a soil-borne disease that commonly occurs in alfalfa-growing areas. The disease was first reported in the USA in 1937 and has since been reported in Canada, New Zealand, Australia, India, Egypt, and Japan, with incidence rates of over 60% in some areas [2,3,4,5,6]. Similarly, Fusarium root rot commonly affects alfalfa production in the northern region of China. *Fusarium* spp. can cause disease throughout the whole growth period of alfalfa. The disease reduces the ability of the plant to fix nitrogen and the quality of alfalfa, greatly shortening its utilization timeframe; it can also cause alfalfa to lose its processing value [7]. Alfalfa consumes stored organic matter during overwintering, and the soluble sugar content of *Fusarium*-infested alfalfa plants becomes significantly reduced, resulting in the number of tillers being reduced in the following year, which seriously affects yield and quality [7]. Additionally, *Fusarium* spp. can survive in the soil for a long time and accumulate yearly, worsening disease severity in alfalfa and leading to outbreaks as the planting years increase.

*Fusarium proliferatum* is a primary pathogenic fungus that causes root rot disease in many plant species, including danshen *(Salvia miltiorrhizae*) [8], rice (*Oryza sativa*) [9], storage onion(*Allium cepa*) [10], and date palm (*Phoenix dactylifera*) [11]. In addition, *F. proliferatum* has been reported to produce many deleterious fungal toxins, including fusarium acid, fumonisins, fusaproliferin, beauvericin, and moniliformin. These compounds pose a serious threat to global crop safety and livestock and human health [12,13,14,15]. Since *F. proliferatum* is an endophytic fungus, it can live within plants and produce large numbers of conidia, which can survive for many years in the soil [16]. When the weather becomes humid and warm, the *F. proliferatum* conidia germinate and spread through atmospheric dust and rain.

Plants have evolved immunity to pathogens in several manners, including PAMP-triggered immunity (PTI) and effector-triggered immunity (ETI) [17,18]. Plants possess pattern-recognition receptors (PRRs) in the cell membrane, and these PRRs recognize pathogen-associated molecular patterns (PAMPs) such as flagellin and chitin. Recognition of PAMP by PRRs activates PTI, which restricts pathogen development. Most pathogens secrete effectors, that is, avirulence (Avr) proteins, into plant cells to suppress PTI, but various resistance (R) genes in plants can detect these effectors. R genes can recognize specific effectors that induce ETI, known as “gene-for-gene resistance” [17]. Applying these resistant varieties is the most effective way to control such diseases.

Fusarium root rot is a multi-infestation disease, which makes it difficult to control and allows it to seriously affect alfalfa production in China [7]; however, due to technological limitations and high difficulty of research, the progress of research on alfalfa root rot has been very slow. To date, research on the disease has mostly involved field investigation, isolation, and identification of pathogens. There are few reports on the regulatory networks that control disease resistance and screening of resistance and susceptibility genes. Currently, only L. Cong has initially explored the response mechanism of alfalfa after inoculation with *F. proliferatum* using two–dimensional gel electrophoresis (2-DE) and MALDI-TOF/TOF; however, this study was not systematic because of the limited number of differentially expressed proteins [19]. Recently, transcriptomic studies have become a more rapid method for gene mining due to the rapid development of omics research and reduced sequencing cost. Transcriptome sequencing technology has been used for global gene expression profiling in many plants, including alfalfa [20,21], and has been used in several host–pathogen interaction studies [20,22,23,24,25,26,27].

Most transcriptome studies of plant–pathogen interactions have been conducted between two species [27,28], between different cultivars of the same species [29,30], or between different genotypes of the same cultivars [21,31]. Few studies have been conducted between different clonal lines [32]. Alfalfa is a highly heterozygous autotetraploid plant, with considerable differences in seed genotypes among the same varieties [33]. Transcriptome analyses of clones propagated by cutting can eliminate the differences in genotypes between cultivars and species.

In this study, we obtained one susceptible and one resistant plant of “AmeriGraze401 + Z” alfalfa through two rounds of screening, and the two clonal lines were propagated by cutting and then sampled for transcriptomic sequencing after inoculation with *F. proliferatum* L1. The analyses of differentially expressed genes (DEGs) focused on the functional classification and the discovery of novel genes that responded to *F. proliferatum* L1, particularly those involved in plant–pathogen interactions. The current study provides new insights into the resistance regulatory mechanisms against *Fusarium* root rot in alfalfa and provides valuable information for breeding alfalfa cultivars that are resistant to *Fusarium* root rot.

## 2. Materials and Methods

### 2.1. Experimental Design

The experimental design is shown in Figure 1. After two rounds of disease resistance screening, one resistant and one susceptible clonal line of alfalfa “AmeriGraze401 + Z” were used for transcriptome sequencing analyses.

### 2.2. Screening of Susceptible and Resistant Alfalfa Plants

Alfalfa is a highly heterozygous cross-pollinated plant, and individual alfalfa plants have different genotypes within the variety. To overcome the differences caused by these differing genotypes, clonal lines of resistant and susceptible individual plants were screened as plant materials for this study. Our previous study [19] indicated that “AmeriGraze401 + Z” was a cultivar that was more resistant to Fusarium root rot based on an evaluation of disease resistance. The pathogen used in this experiment was *F. proliferatum* strain L1, which was collected and preserved in our laboratory [34]. Two rounds of screening were conducted to ensure the accuracy of screening resistant and susceptible individual plants. The first round entailed a preliminary soil culture screening method [35], followed by hydroponics for the second round of screening [36]. For the first round of screening, alfalfa seeds were sterilized in 25% bleach (*v/v*) for 15 min, followed by rinsing three times with sterilized water. Next, 1000 sterilized seeds were planted in a sterile soil mixture (soil:vermiculite:perlite = 3:2:1 (*w/w/w*)) with five seeds per pot. After the alfalfa seeds grew for 30 d, they were inoculated with sterile sorghum (*Sorghum bicolor*) seeds containing *F. proliferatum* and cultivated in a greenhouse (25 ± 2 °C day/20 ± 2 °C night, with 75–80% relative humidity, 16 h light/8 h dark). After inoculation 45 d, resistant plants with small root disease spots and susceptible plants with heavy root rot were screened and then propagated using the hydroponic cutting method [37]. After cutting, the plants were rooted for 20 d in a growth chamber under controlled conditions (25 ± 1 °C day/20 ± 1 °C night, with 80% relative humidity and 16 h light (200 mol/m^2^s)/8 h dark). Uniform clonal line plants were inoculated with 5 × 10^6^ spores/mL of *F. proliferatum* L1 under hydroponic cultivation. After 14 d, one of the most resistant and one of the most susceptible plants were selected and propagated using the hydroponic cutting method [37]. The two clonal lines were used for transcriptome sequencing analyses.

### 2.3. Determination of Sampling Time and Sample Collection for RNA-Seq

In order to explore alfalfa early response genes during compatible interaction with *F. proliferatum* L1, we examined the expression of defense-related genes at different time points after inoculation using real-time quantitative reverse-transcription PCR (qRT-PCR).

*NPR1* and *NPR3* have been identified as important disease-resistant-related genes in *Arabidopsis thaliana* [38]. Additionally, *NPR1* plays a significant role in the establishment of systemic acquired resistance (SAR) as well as induced systemic resistance (ISR). *SPL15* is an *IPA1* homologous gene and has been supposed as an upstream disease-resistant transcription factor in the disease resistance of rice [39]. In our previous study, we also verified that *SPL15* participates in the process of alfalfa root rot resistance (unpublished data). Therefore, the relative expression levels (2^−∆∆t^) of the disease-resistant genes, *NPR1* and *NPR3*, and their upstream transcription factor, *SPL15*, were determined by qRT-PCR at 0, 12, 24, 48, 72 h, and 7 d after inoculation with *F. proliferatum* L1. The inoculation method refers to the hydroponic inoculation previously established by our laboratory [34]. According to the expression level of *NPR1*, *NPR3,* and *SPL15* genes, the sampling time was determined for transcriptome sequencing.

The two clonal lines prepared in Section 2.2 were inoculated with *F. proliferatum* L1 for transcriptome sequencing analyses with three replicates. The whole roots of the inoculated and uninoculated (control) groups were collected, frozen in liquid nitrogen, and stored at −80 °C for transcriptome sequencing.

### 2.4. Transcriptome Sequencing

A total of 48 samples at four time points (12 samples of uninoculated resistant clonal line, 12 samples of inoculated resistant clonal line, 12 samples of uninoculated susceptible clonal line, and 12 samples of inoculated susceptible clonal line) were sent to Personalbio Co., Ltd., (Shanghai, China) for transcriptome sequencing. Total RNA was extracted from the tissues using TRIzolR^®^ reagent (Plant RNA Purification Reagent for Plant Tissue; Invitrogen Corp., Carlsbad, CA, USA). The sequencing libraries were constructed using the VAHTS mRNA-seq V3 Library Prep Kit from Illumina (San Diego, CA, USA) with 1 µg of high-quality total RNA (A260/A280 = 1.8–2.2, A260/A230 ≥ 2.0, RNA integrity number (RIN) ≥ 6.5, 28S:18S ratio ≥ 1.0, >2 µg samples). The sequencing library was then sequenced using an Illumina HiSeq xten/NovaSeq 6000 sequencer.

Raw sequences were qualitatively controlled (short fragments and low-quality fragments were removed) to obtain high–quality clean sequences. The *M. sativa* cv. Xinjiangdaye genome was used as the reference sequence [40]. To ensure precise alignment, the raw RNA sequence data were removed with a 3-end adapter using Cutadapt (https://cutadapt.readthedocs.io/en/stable/ (accessed on 1 January 2022)), and then reads with average quality below Q20 and minimum read size (50 bp) were filtered out. The filtered reads were used for sequence alignment to the reference genome using HISAT2 software (https://daehwankimlab.github.io/hisat2/ (accessed on 3 April 2022)). The read count value was determined by HTSeq (https://htseq.readthedocs.io/en/master/ (accessed on 24 February 2022)).

### 2.5. Differential Gene Expression Analysis of Alfalfa

In order to better exploit resistance-related genes against Fusarium root rot in alfalfa, we analyzed four types of DEGs: (1) the DEGs between resistant (R) and susceptible (S) clonal lines before inoculation (control, C) at each time point (CR24 vs. CS24, CR48 vs. CS48, CR72 vs. CS72, CR7d vs. CS7d); (2) the DEGs between uninoculated (C) and inoculated (Treatment, T) of resistance clonal line at each time point (CR24 vs. TR24, CR48 vs. TR48, CR72 vs. TR72, CR7d vs. TR7d); (3) the DEGs between uninoculated and inoculated groups of susceptible clonal lines at each time point (CS24 vs. TS24, CS48 vs. TS48, CS72 vs. TS72, CS7d vs. TS7d); (4) the DEGs between resistance and susceptible clonal lines after inoculation at each time point (TR24 vs. TS24; TR48 vs. TS48; TR72 vs. TS72, TR7d vs. TS7d). DESeq 2 (DESeq2 R package 1.16.1) was used to screen the DEGs, and the screening conditions were *p*-adjusted < 0.05 and |log_2_FC| ≥ 1. The differential gene expression sequences were annotated by NR, IPR, TREMBL, and Swiss-Prot based on the reference genome. Clonal line type-specific DEGs were obtained by removing the genetically different genes DEGs generated in uninoculated group from all the DEGs induced by *F. proliferatum* L1 inoculation using FunRich software (http://www.funrich.org/ (accessed on 6 March 2022)). Gene ontology (GO) enrichment analysis of the DEGs was performed using topGO, and Kyoto Encyclopedia of Genes and Genomes (KEGG) enrichment analyses (*p*-value ≤ 0.05) were conducted using clusterProfiler according to biological process (BP), molecular function (MF), and cellular component (CC) among all up- and downregulated genes at each time point after inoculation.

### 2.6. Differential Gene Expression Analysis of F. proliferatum L1

The number and expression level of *F. proliferatum* L1 genes can be detected in the inoculation group of the two clonal lines. The *F. proliferatum* ET1 genome was used as the reference sequence for sequence alignment, transcriptome, and alignment. GO and KEGG enrichment analyses were performed on detected genes.

### 2.7. Real-Time Quantitative PCR (qRT-PCR) Analysis of the DEGs

In order to verify the expression patterns of genes observed in the RNA-seq analysis, we used the total RNA of all the samples subjected to transcriptome sequencing for qRT-PCR verification. A 1 μg sample of total RNA was reverse-transcribed into cDNAs using the PrimeScript^TM^RT reagent kit with gDNA Eraser (TaKaRa, Kusatsu, Japan). A NanoDrop One (Thermo Fisher Scientific Inc., Waltham, MA, USA) was used to quantify 100 ng of purified single–stranded cDNA for qRT–PCR. Relative quantitative analyses were performed in a CFX-96 Touch Real-time PCR detection system (Bio-Rad, Inc., Hercules, CA, USA) with the following cycling conditions: 95 °C for 30 s and 40 cycles at 95 °C for 5 s, 60 °C for 30 s.

Three technical replicates were performed for each sample. Gene-specific primers were designed using Primer Premier 5 software (PREMIER Biosoft, San Francisco, USA) (Appendix A). Relative quantification was normalized to the housekeeping control gene (*β*-actin), and the fold change (FC) in gene expression was calculated using the 2^−ΔΔt^ method.

## 3. Results

### 3.1. Screening of Experimental Plants and Determination of Sampling Time

In the first screening, 58 susceptible and 42 resistant plants with obvious traits were selected for the second screening (Figure 2A,B). After propagation by the hydroponic cutting method and *F. proliferatum* L1 inoculation under hydroponic conditions, one of the most resistant and one of the most susceptible clonal lines (Figure 2C,D) with obvious symptoms were screened after surface sterilization to generate uniform clonal lines by cutting propagation for transcriptome sequencing.

Finally, qRT–PCR was performed to determine the relative expression levels (2^−∆∆t^) of the disease-resistance-related genes, *NPR1* and *NPR3*, and their upstream transcription factor *SPL15* at six time points. The expression levels of *NPR1* and *SPL15* were significantly increased in the inoculated group compared to those in the uninoculated group, although the *NPR3* gene was only increased slightly at 24 h (Figure 2E). No obvious disease symptoms were found after inoculation until 7 d, when the plants exhibited yellowing or falling (Figure 2F). Based on the qRT–PCR results and post-inoculation phenotype, we collected samples at 24 h post-inoculation (hpi), 48 hpi, 72 hpi, and 7 days post-inoculation (dpi) for transcriptome sequencing.

### 3.2. Alfalfa Sequence Analysis and Alignment with the Reference Genome

For each sample, approximately 8 Gb of clean sequences were obtained. The clean reads% and clean data% for the library transcriptome sequencing obtained from uninoculated or inoculated resistant and susceptible clonal lines were both higher than 90% (Appendix A).

On average, 85% of the total sequences were mapped to the *M. sativa* cv. Xinjiangdaye reference genome sequence, with most of the sequences being mapped to exons—approximately 96%—and only a small portion of them being mapped to intergenic regions, approximately 17% (Appendix A). To evaluate the similarity of RNA-Seq data from different samples, the mapped RNA-Seq data from different samples were used for a principal component analysis (PCA) with DEseq2 (Appendix A). In addition, Pearson’s correlation among biological replicates (Appendix A) for all the samples was analyzed and showed a high correlation between sequencing replicates (approximately 0.95). The results of PCA and Pearson’s correlation indicated there is a high correlation between sequencing replicates.

### 3.3. DEGs between Resistant and Susceptible Clonal Lines in Uninoculated or Inoculated Conditions

The numbers of DEGs between resistant and susceptible clonal lines in uninoculated or inoculated conditions were analyzed at four time points (Figure 3A).

To eliminate the effects of genetic differences, DEGs between resistant and susceptible clonal lines at the same time points were identified before inoculation (CR 24 vs. CS24, CR48 vs. CS48, CR72 vs. CS72, and CR7d vs. CS7d) (|log_2_FC| ≥ 1 and FDR < 0.05). There were 1057 overlapped DEGs detected at four time points (Figure 3B). Among the 1057 DEGs, the gene expression level of 592 DEGs at 24 hpi, 748 DEGs at 48 hpi, 741 DEGs at 72 hpi, and 607 DEGs at 7 dpi in resistant plants were higher than that in susceptible clonal line. Additionally, the expression levels of 380 DEGs in the resistant clonal line were always higher than that in the susceptible clonal line at four time points, but the expression levels of only 237 DEGs in the susceptible clonal line were higher than that in the resistant clonal line. In addition, more DEGs were detected between the resistant and susceptible clonal lines at 24 h (CR24 vs. CS24). GO enrichment analysis and KEGG enrichment analysis were conducted for these 1057 DEGs, and these genes were enriched in 712 biological processes and 86 KEGG pathways (Appendix A). Among the 712 biological processes, the top five biological processes with more DEGs were biological, metabolic, cellular, organic substance metabolic, and primary metabolic processes. Among the 86 KEGG pathways, the top five KEGG pathways with more DEGs were metabolic, biosynthesis of secondary metabolites, ribosome, carbon metabolism, and biosynthesis of amino acids pathways. We hypothesized that there might be differences in the existence of structural, physical, or chemical barriers between resistant and susceptible clonal lines based on DEG analysis before inoculation (CS vs. CR).

Among the 1057 DEGs, we focus on the genes enriched in biological processes related to defense, such as the response to stress, cell wall composition, and the genes enriched in defense-related pathways, such as the biosynthesis of secondary metabolites, plant–pathogen interaction, plant hormone signal transduction, and the MAPK signaling pathway.

A total of 27 DEGs among the 1057 DEGs were enriched in biological processes related to defense. We detected 11 genes of the 27DEGs associated with the cell wall composition, as follows: *MS.gene022147*, *MS.gene027454*, *MS.gene56020*, *MS.gene88663*, *MS.gene014219*, *MS.gene013427*, *MS.gene045626*, *MS.gene063617*, *MS.gene06527*, *MS.gene46272*, and *MS. gene98660*. *MS.gene022147* and *MS.gene56020* are annotated as chitinase-like proteins, *MS.gene02745*4 and *MS.gene88663* are annotated as xyloglucan endotransglucosylase/hydrolase proteins, and the expression level of these four genes is higher in the resistant line than in the susceptible line. *MS.gene014219* is annotated as 5′-methylthioadenosine nucleosidase, a broad-spectrum antimicrobial drug target, and it was highly expressed in the resistant lines compared to the susceptible lines. The other six genes (*MS.gene013427*, *MS.gene 045626*, *MS.gene063617*, *MS.gene06527*, *MS.gene46272*, and *MS.gene98660*) are annotated as expansin-like proteins. The expression level of *MS.gene013427*, *MS.gene045626*, and *MS.gene98660* is higher in the resistant line than in the susceptible line. However, the expression level of *MS.gene063617* and *MS.gene46272* is higher in the susceptible line than in the resistant line. The expression level of *MS.gene06527* is higher in the resistant line than in the susceptible line at the 24 h time point while lower in the susceptible line than in the resistant line at other time points. There were 16 DEGs among 27 DEGs related to response to stress (Appendix A), and the gene expression levels of 2 genes (*MS.gene029046* and *MS.gene000621*) were always higher in resistant lines than susceptible lines at four time points, and the gene expression levels of 6 genes (*MS.gene004925*, *MS.gene42084*, *MS.gene82245*, *MS.gene008075*, *MS.gene54691*, and *MS.gene71844*) were always higher in the susceptible line than the resistant line at four time points.

There were more DEGs identified in the inoculated groups (TR 24 vs. TS24, TR48 vs. TS48, TR72 vs. TS72, and TR7d vs. TS7d) than in the uninoculated group (CR 24 vs. CS24, CR48 vs. CS48, CR72 vs. CS72, and CR7d vs. CS7d), and a total of 67,612 DEGs between the two inoculated lines were identified at four time points (24 hpi, 48 hpi, 72 hpi, and 7 dpi). More DEGs between the inoculated resistant line and susceptible line were identified at 48 hpi (23,540), followed by 7 dpi (23,299), 24 hpi (20,993), and 72 hpi (20,773). A total of 10,926 DEGs overlapped at four time points, and the number of overlapped DEGs at all four time points was approximately 10-fold higher in the inoculated group than in the uninoculated group (Figure 3C). More DEGs were detected between the resistant and susceptible clonal lines at 48 h (TR48 vs. TS48), and the top five biological processes with more DEGs were structural molecule activity, structural constituent of ribosome, small molecule binding, nucleotide binding, and nucleoside phosphate binding. Similarly, it was found that GO terms and KEGG of the top 20 observed in the TR48 vs. TS48 were almost the same as in the CR24 vs. CS24 pair (Appendix A).

### 3.4. Transcriptional Changes in Response to F. proliferatum L1 Inoculation

Resistant and susceptible clonal lines inoculated with *F. proliferatum* L1 were compared with the uninoculated group, and the DEGs that responded to the pathogen infection in each clonal line at four time points were obtained (|log_2_FC| ≥ 1 and FDR < 0.05) (Appendix A, Figure 4).

Compared with the uninoculated group, the DEGs of the resistant line consisted of 4475 downregulated and 5127 upregulated genes (24 hpi), 25 downregulated and 7 upregulated genes (48 hpi), 1139 downregulated and 1040 upregulated genes (72 hpi), and 12,332 downregulated and 13,818 upregulated genes (7 dpi). In the susceptible clonal line, downregulated DEGs between the uninoculated and inoculated groups were 14,995, and upregulated were 12,148 at four tome points. The DEGs of the susceptible line consisted of 9136 downregulated and 9349 upregulated genes (24 hpi), 208 downregulated and 468 upregulated genes (48 hpi), 6490 downregulated and 5944 upregulated genes (72 hpi), and 8505 downregulated and 6204 upregulated genes (7 dpi).

The number of DEGs between the inoculated and uninoculated resistant clonal lines was much less than that of the susceptible clonal line at the early stage of inoculation (24, 48, and 72 hpi) (Figure 4A,C). In particular, there were more DEGs in the susceptible line in the early stage (24 h), but there were more DEGs in the resistant line at the later stage (7 dpi). We analyzed the top 30 genes with a high fold change between uninoculated and inoculated groups of each line at four time points (Appendix A). The top 30 genes with a high fold change compared with those of the control in the two clonal lines at 24 hpi and 7 dpi are shown in Figure 4. At 24 h, we found that 19 genes of the 30 genes were upregulated in the resistant line, while all the genes were downregulated in the susceptible line. Additionally, there was no overlap in the DEGs between the two lines. GO enrichment analyses of these DEGs indicated enrichment in small molecule binding, catalytic activity, nucleotide binding, nucleoside phosphate binding, and anion binding. KEGG enrichment analyses of these DEGs indicated enrichment in glycolysis/gluconeogenesis, citrate cycle (TCA cycle), fatty acid degradation, pyruvate metabolism, alanine, aspartate and glutamate metabolism, phenylpropanoid biosynthesis, caffeine metabolism, and the pentose phosphate pathway (Appendix A). Notably, in the resistant line, an interesting phenomenon was observed: the miraculin genes (*MS.gene37740*, *MS.gene57814*, *MS.gene57819*, *MS.gene37741*, *MS.gene377 37*, *MS.gene069488*, *MS.gene57815*, and *MS.gene 069445*) were the most highly expressed. Moreover, a large number of polygalacturonase-inhibitor genes (*MS.gene41336*, *MS.gene 34631*, *MS.gene41335*, *MS.gene43357,* and *MS.gene05228*) were upregulated; however, heat shock protein 70 (HSP 70, *MS.gene25760*) and arabinogalactan protein 14 (*MS.gene08201*) were found to be downregulated. In the susceptible line, more genes participate in metabolic, cellular, and single-organism processes, such as cytochrome P450 (*MS.gene57146*), peroxidase (*MS.gene071284*), and glutathione S-transferase (*MS.gene055150*). At 7d, 16 genes of the top 30 genes were upregulated in the resistant line, while 29 genes were downregulated, and one gene (*MS.gene014747*) was upregulated in the susceptible line. In the resistant line, the DEGs were mainly classified into cellular and metabolic processes, such as chalcone synthase (*MS.gene024774*), ferredoxin (*MS.gene76234*), and potassium transporter (*MS.gene61263*) genes. Disease-resistant genes were also detected, such as the F-box/FBD/LRR-repeat protein gene (*MS.gene98017*) and disease-resistant protein gene (*MS.gene73019*). In the susceptible line, senescence-associated carboxylesterase, glutamine synthetase, threonine synthase, and electron transfer flavoprotein–ubiquinone oxidoreductase (*MS.gene98729*) genes were downregulated.

### 3.5. Common Transcriptional Changes in Response to the Inoculation of Resistant and Susceptible Clonal Lines

A total of 2078 overlapping DEGs between the uninoculated and inoculated groups were identified in both the clonal lines (Figure 5, Appendix A). We performed a functional enrichment analysis for GO terms and found that most of these genes were enriched in the polysaccharide catabolic process, polysaccharide biosynthetic process, reproduction, and the response to acid chemical processes (Appendix A). KEGG enrichment analyses were performed for these DEGs, and after excluding genes with unspecified functions, common genes related to metabolic pathways (50.4%) accounted for the largest proportion, followed by genes related to plant–pathogen interactions (43%), including the categories: biosynthesis of secondary metabolites (29.7%), plant hormone signal transduction (5.9%), phenylpropanoid biosynthesis (5.1%), plant–pathogen interaction (4.3%), and the MAPK signaling pathway (3.1%) (Appendix A).

There were many transcription factors (TFs) in these DEGs that were shown to be involved in resistance to pathogens. They included bHLH (*BHLHW*, *BHLH123*, *BHLH30*, *BBD1, BHLH25*, *BHLH80*, and *BHLH126*), AP2 (*BBM2*, *ANT*, and *WRI1*), SBP (*ALPL* and *SPL6*), WRKY (*WRKY23*, *WRKY75*, *WRKY6*, *WRKY51*, *WRKY48*, *WRKY70*, *WRKY40*, *WRKY1*, and *WRKY42*), and MYB (*MYB87*, *MYB78*, *MYB108*, *MYB88*, and *MYB4*) TF families. These plant–pathogen interaction genes and disease-resistant-related transcription factors showed roughly the same trend at the four time points in both clonal lines after inoculation compared to the uninoculated plants (Figure 6).

### 3.6. Clonal Line Type-Specific Transcriptional Changes in Response to Inoculation

After removing the DEGs of the genetically different genes that were generated in the uninoculated group from all the DEGs induced by *F. proliferatum* L1 inoculation (Figure 4A,C) using FunRich software, the remaining DEGs were treated as transcriptional changes in response to *F. proliferatum* L1 inoculation at four time points (Figure 5, Appendix A). Specific transcriptional changes induced by *F. proliferatum* L1 included 3512 and 4081 genes found to be differentially regulated in the resistant and susceptible clonal lines, respectively (Figure 5). The most prevalent functional categories among the modulated genes were those of metabolic process (41.5% and 43.1%, respectively) and the biosynthesis of secondary metabolites (27.4% and 24.5%, respectively) for both clonal lines. Other categories of plant–pathogen interaction genes included plant–pathogen interactions (5.6% and 6.9%, respectively) and phenylpropanoid biosynthesis (4.7% and 3.3%, respectively). Additionally, plant hormone signal transduction (3.7% and 5.6%, respectively) and the MAPK signaling pathway (2.6% and 4.4%, respectively) were prevalent in both clonal lines. These percentages were approximately consistent with the percentage of DEGs associated with plant–pathogen interactions found in the common transcriptional changes in both clonal lines (Appendix A).

In the resistant clonal line, 485 clonal line type-specific DEGs were detected at 24 hpi (Figure 7). Among these genes, 212 were downregulated, and 273 were upregulated. Six genes (*MS.gene65772*, *MS.gene38785*, *MS.gene067433*, *MS.gene011203*, *MS.gene00935*, and *MS.gene 055695*) of the top ten DEGs with a high fold change were upregulated. These six genes were annotated as a transmembrane protein, methyltransferase DDB, protein nuclear fusion defective 4 isoform X1, cytochrome P450 88D2, epidermal patterning factor protein 2, and receptor-like protein 18, respectively. Four genes (*MS.gene98474*, *MS.gene013794*, *MS.gene 038488*, and *MS.gene71722*) of the top ten DEGs were downregulated. These four genes were annotated as LRR and NB-ARC domain disease-resistance protein, metal transporter Nramp5, P-loop nucleoside triphosphate hydrolase superfamily protein, and transmembrane protein, respectively. In susceptible clonal line, five genes, including *MS.gene26169* (protein reveille 1), *MS.gene43364* (polygalacturonase inhibitor), *MS.gene37066* (MYB family transcription factor PHL5), *MS.gene08790* (plasmodesmata callose-binding protein 3), and *MS.gene66665* (aluminum-activated malate transporter 2 isoform X1), were upregulated at all four time points. Additionally, TF families, including bHLH, AP2, MYB, WRKY, and SBP, were analyzed in all the DEGs induced by *F. proliferatum* L1 in the resistant and susceptible clonal lines, respectively. These TF families exhibited different expression trends in specific and common DEGs, but these DEGs changed significantly during the early stage in the susceptible clonal line and changed significantly during the late stage in the resistant clonal line (Appendix A).

### 3.7. F. proliferatum L1 Genes Were Enriched in Resistant and Susceptible Clonal Lines after Inoculation

Using *F. proliferatum* ET1 as the reference genome, the number of fungal genes detected in the inoculation groups was counted. GO and KEGG enrichment analyses were performed on these genes, and the results indicated that they were mostly low-molecular-weight secondary metabolites such as lipids, glycosides, polysaccharides, peptides, proteins, and glycoproteins, which are typical pathogenic factors of host-selective fungi. The patterns of expression of fungal genes were distinct during the interactions between the two lines. We found that the number of fungal genes detected in the susceptible clonal line was significantly higher (one–way analysis of variance (ANOVA), *p* < 0.001) than that detected in the resistant clonal line during the early stage. Although the number of fungal genes detected in the susceptible line at 24 h was slightly higher than that of the resistant line, they were consistent at the other time points, and the top 30 genes expressed at the highest levels detected in the two lines at the four time points were basically consistent (Appendix A, Figure 8).

### 3.8. Validation of RNA-Seq Data by qRT-PCR

To validate the RNA–Seq expression profiles of DEGs, qRT–PCR was performed on three disease–associated genes (*NPR1*, *PR1*, and *PR4A*), eight disease–associated transcription factors (*WRKY22*, *WRKY29*, *WRKY33*, *SPL6*, *SPL7*, *SPL9*, *SPL15*, and *SPL16*), and *MS.gene03877* (NBS–LRRtype disease resistance protein). For the relative expression levels of these 12 genes in the resistant and susceptible clonal lines and inoculated relative to the uninoculated group at four time points, the qRT–PCR results were approximately consistent with those of the RNA–Seq data. However, the RNA–Seq analysis displayed a higher dynamic range, showing larger differences between the fold change compared with qRT–PCR (Figure 9).

## 4. Discussion

In this study, we generated a unique set of DEGs induced by *F. proliferatum* L1 in alfalfa. Overall, 85% of the total sequences were mapped to the *M. sativa* cv. Xinjiangdaye reference genome sequence. This finding is significant for understanding the temporal expression of genes in alfalfa after inoculation with *F. proliferatum* L1. It is also can be used as a candidate gene pool against *Fusarium* spp.

The number of DEGs in the susceptible clonal line between the inoculated and uninoculated groups was twice that of the resistant clonal line between inoculated and uninoculated groups at 24 hpi. Many of these DEGs overlapped between the two uninoculated clonal lines. We hypothesize that this may be due to the genetic differences between the two clonal lines in their physical structure and chemical barrier against *F. proliferatum*. In our study, 17 respiratory burst oxidase homolog protein (RBOH) genes were detected. After inoculation, the change trends of these 17 genes in the two clonal lines were basically the same, but the differential expression in the susceptible clonal line occurred earlier than that in the resistant clonal line, possibly due to the pathogen invading the susceptible clonal line earlier and initiating oxidative activity. Notably, the miraculin gene was the most highly expressed gene at 24 h after inoculation in the resistant line. Previous researchers have focused on the benefits of miraculin to the human and have not reported on the function of miraculin on the plant itself [41,42]; however, a new study showed that miraculin might primarily play a role in regulating seed germination and maturation, resisting pathogen infection and environmental stress, and regulating plant growth [43]. These results indicate that the peculiar property of miraculin that modifies sour tastes to sweet tastes may be secondary, and the main meaning of its existence is to benefit itself. A total of eight *HSP90* genes were detected, among which three DEGs (*MS.gene30654, MS.gene38210, and MS.gene49154*) were shared in two clonal lines after inoculation but were expressed earlier in the resistant line than in the susceptible line. Three DEGs (*MS.gene24136, MS.gene58998, and MS.gene81285*) were unique in the resistant line, and two DEGs (*MS.gene80872* and *MS.gene81288*) were unique to the susceptible line after inoculation (Appendix A). These results suggest that the early expression of *HSP90* may help plants to remove ROS in the resistant line.

The difference in the plant immunity between the two clonal lines may also be the reason for their different defense response levels to the pathogens. Plants have evolved immunity to pathogens via either PTI or ETI [17,18]. Plants possess PRRs on their surface membrane that trigger PTI by PAMPs and belong to families of receptor–like kinases (RLKs) [44,45]. According to KEGG analyses, the DEGs of LRR receptor–like serine/threonine–protein kinase FLS2, LRR receptor–like serine/threonine-protein kinase EFR, receptor kinase–like protein Xa21, probable LRR receptor-like serine/threonine-protein kinase, and putative receptor–like protein kinase were involved in the alfalfa immunity. Additionally, some leucine-rich repeats (LRR) receptor–like serine/threonine-protein kinases were up- or downregulated after inoculation in one or both clonal lines. After inoculation, the expression of these genes in the resistant clonal line was significantly upregulated compared to that in the uninoculated group, whereas those in the susceptible clonal line were almost unchanged (Appendix A).

Downstream signaling networks are triggered by pathogen recognition and mediated by protein kinases (PK), particularly calcium–activated protein kinases (CIPKs or CDPK) and mitogen–activated protein kinase (MAPK), which control defense responses [46,47,48]. Calcium-dependent protein kinases (CDPKs) comprise a large family of serine/threonine kinases in plants and protozoans [48]. In our data, no CDPK genes were detected in the common transcriptional changes in response to inoculation and in the DEGs specific to the resistant clonal line, whereas three CDPK genes were detected in the DEGs specific to the susceptible clonal line, compared with the inoculation group. The three CDPK genes (*MS. gene41297*, *MS.gene50367*, and *MS.gene60795*) increased slightly during the early stage and changed significantly at 72 h. The expression of *MS. gene41297* decreased, while those of *MS.gene50367* and *MS.gene60795* increased (Appendix A). MAPK cascades are highly conserved signaling modules downstream of receptors/sensors that transduce extracellular stimuli into intracellular responses in eukaryotes. Plant MAPK cascades play pivotal roles in signaling plant defenses against pathogen attacks. MAPK cascades have also emerged as battlegrounds for plant–pathogen interactions. In Arabidopsis, pathogen perception by plants leads to the activation of two independent MAPK cascades [49,50]: the MEKK1–MKK4/MKK5–MPK3MPK6 cascade and the MEKK1-MKK1/MKK2-MPK4 cascade. Two *MEKK1* genes (*MS.gene73935* and *MS.gene94111*) were detected in the DEGs of the resistant clonal line after inoculation. Both genes were slightly upregulated or downregulated in the early stages. However, both were significantly upregulated at 7 d.

In our study, we found that 18 *CERK1* genes were expressed, and the expression levels of these genes were basically the same in the two clonal lines without inoculation, but expression levels in the resistant clonal line were slightly higher than those in the susceptible clonal line. The expression level of these genes slightly increased in the inoculated resistant clonal line compared to the uninoculated resistant clonal line, whereas there was a decrease in the susceptible clonal line during the early (24 hpi) and late (7 dpi) stages after inoculation and an increase in the middle stages (48 hpi and 72 hpi) (Appendix A).

Downstream defense-responsive genes are normally positively or negatively regulated by TFs and are direct or indirect targets of various signal-transduction pathways. The TF families found in our study are widely reported to be involved in plant defense responses, including MYB, WRKY, ethylene-responsive factors (ERFs), squamosa promoter-binding protein–like (SPL), zinc finger domain proteins, and basic helix–loophelix (bHLH). The basic helix–loop–helix (bHLH) transcription factor family is one of the largest transcription factor gene families in *Arabidopsis thaliana*, and these transcription factors have the pleiotropic regulatory roles in plant growth and development, stress response, biochemical functions, and the web of signaling networks [51]. These common disease-resistance-related transcription factors in response to inoculation showed approximately the same trend at the four time points in both lines between the inoculated and uninoculated groups. However, type–specific differentially expressed TFs of changes in the clonal line in response to inoculation were less in the resistant clonal line than in the susceptible clonal line, but there was greater variation in the resistant clonal line.

The phenylpropanoid pathway generates secondary metabolites such as lignin, flavonoids, and phytoalexins, which are involved in plant defense against pathogens [52]. Flavonoid biosynthesis begins with the amino acid phenylalanine, and the terminal products include anthocyanins, flavones, isoflavones, and condensed tannins. Phytoalexins such as pisatin and lignans are well-known defense metabolites because of their potent anti-fungal activities and their ability to inhibit secreted fungal enzymes [53]. In our study, secondary metabolism was the category with the largest number of enriched genes in addition to metabolic processes (Appendix A). Biosynthesis of phenylpropanoids (*MS.gene71 342*, *MS.gene29407*, *MS.gene08187*, *MS.gene44869*, etc.), flavonoids (*MS.gene75693*, *MS.gene 89654*, and *MS.gene02599*), lignin (*MS.gene04038* and *MS.gene66786*), and anthocyanins (*MS.gene00229*, *MS.gene05556*, and *MS.gene02621*) were detected in both clonal lines after inoculation (Appendix A).

The *Fusarium* genus contains filamentous ascomycete fungi that can infect a diverse range of plants, and a large number of similar phytopathogenic genes undergo diverse selection during host–pathogen interactions [54]. Evolution has equipped *Fusarium* pathogens with a wide variety of infection strategies [55]. These include the production and secretion of proteins and other effectors to successfully facilitate the infection process by reprogramming the host metabolism and parasitic colonization by manipulating the host cell’s immune response [56]. In the inoculation group, we detected the genes related to lipids (*gene-FPRO_00503* and *gene-FPRO_01999*), glycosides, polysaccharides, peptides (*gene-FPRO_03131*), and proteins (*gene-FPRO_00137*, *gene-FPRO_13465,* and *gene-FPRO_ 03476*) of the fungus that recognized the different secreted effectors of *Fusarium* pathogens (Appendix A).

## 5. Conclusions

To the best of our knowledge, this is the first study to systematically study the defense transcriptome and mining-resistant-related genes response to *Fusarium* spp. infection in alfalfa. In summary, DEGs were identified in control samples at 24 h, 48 h, 72 h, and 7 d after inoculation with *F. proliferatum* L1 in resistant and susceptible clonal lines of alfalfa. Importantly, we studied plant–pathogen interaction genes, plant hormone signal transduction, the MAPK signaling pathway, secondary metabolism, multiple disease resistance proteins, TFs, and genes involved in cell wall expansion and antioxidant processes that were modulated by inoculation with *F. proliferatum*L1. Overall, this study extended our understanding of the d molecular defense of two clonal lines with different genetic backgrounds during *F. proliferatum* inoculation. We identified several candidate genes that could be useful for future research and expect that these data will provide valuable information for research on alfalfa root rot.

## Figures and Tables

**Figure 1 genes-13-00788-f001:**
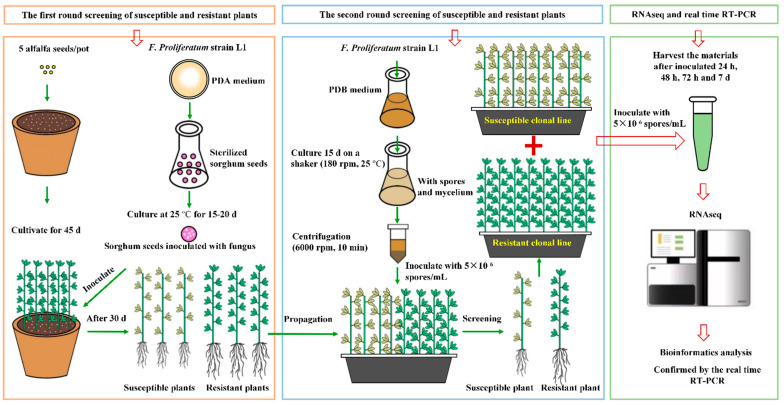
Flow chart of the experimental design.

**Figure 2 genes-13-00788-f002:**
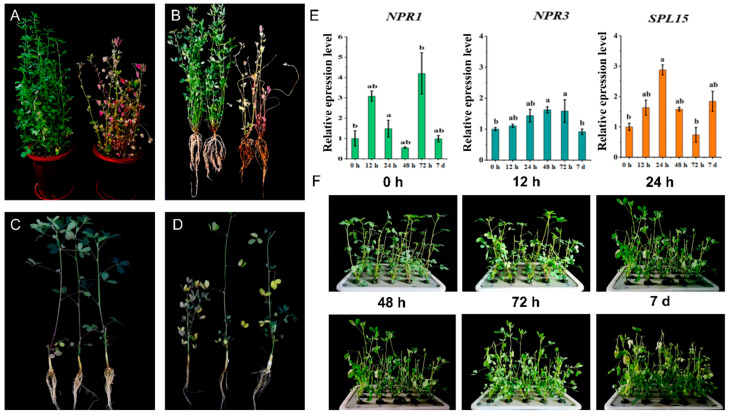
Screening of experimental plants and determination of the sampling time. (**A**,**B**) First screening of resistant (left) and susceptible plants (right). (**C**,**D**) Second screening of resistant (**C**) and susceptible plants (**D**). (**E**) The relative expression levels of the *NPR1*, *NPR3*, and *SPL15* at 24 h, 48 h, 72 h, and 7 d. (**F**) Phenotype of the mixture of susceptible and resistant clonal lines at 24 h, 48 h, 72 h, and 7 d after inoculation. The different letters in (**E**) indicate significant differences in expression levels of *NPR1*, *NPR3,* and *SPL15* in inoculated group compared to those in uninoculated group (*p* < 0.05).

**Figure 3 genes-13-00788-f003:**
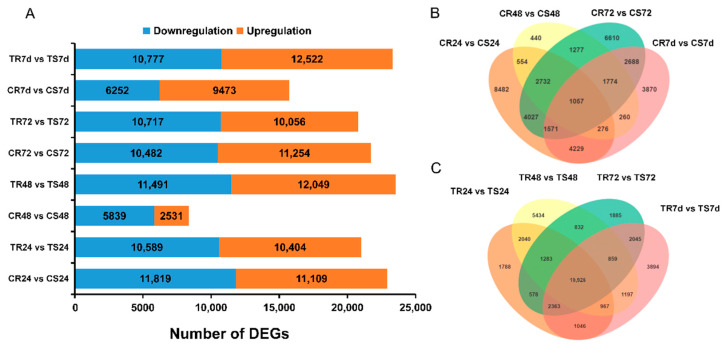
DEGs of uninoculated and inoculated groups in two lines. (**A**) The number of up- and downregulated DEGs between the uninoculated and inoculated groups. (**B**) A Venn analysis of DEGs between control groups at four time points. (**C**) Venn analysis of differentially expressed genes between inoculated groups at four time points. DEGs: differentially expressed genes; CS: uninoculated groups of susceptible clonal line; TS: inoculated groups of susceptible clonal line; CR: uninoculated groups of resistant clonal line; TR: inoculated groups of resistant clonal line.

**Figure 4 genes-13-00788-f004:**
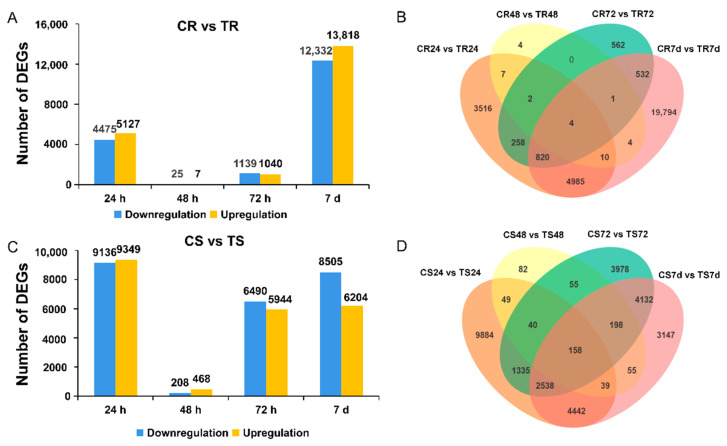
Transcriptional changes in response to F. proliferatum L1 inoculation (|log_2_FC| ≥ 1 and FDR < 0.05). (**A**) The number of DEGs was up- or downregulated at the four time points after inoculation of resistant clonal line. (**B**) Venn plots of the DEGs at the four time points after inoculation of resistant clonal line. (**C**) The number of DEGs was up- or downregulated at the four time points after inoculation of susceptible clonal line. (**D**) Venn plots of DEGs at the four time points after inoculation of susceptible clonal line. DEGs, differentially expressed genes.

**Figure 5 genes-13-00788-f005:**
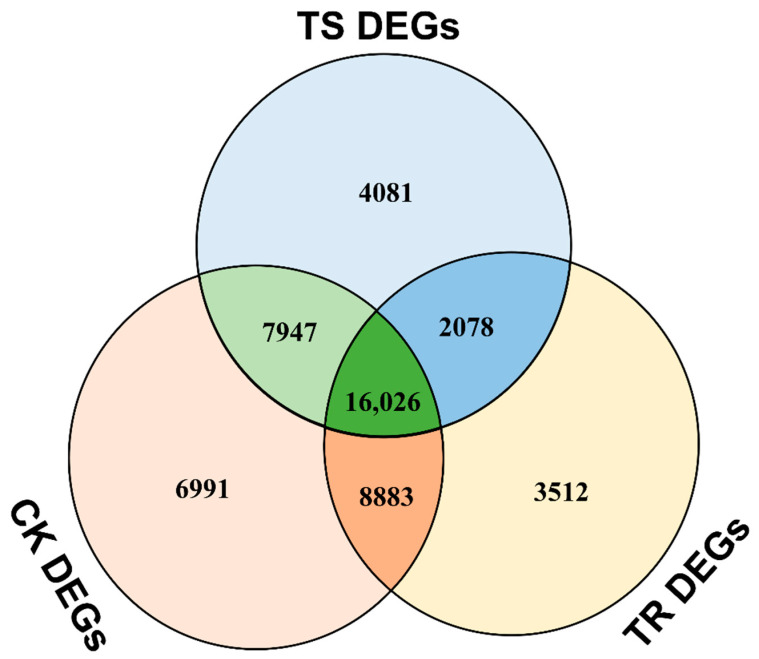
Venn diagrams of CK DEGs, TS DEGs and TR DEGs. CK: all DEGs between uninoculated resistant clonal line and the susceptible clonal line at four time points; TS: all DEGs between the uninoculated and inoculated groups of susceptible clonal line at four time points; TR: all DEGs between the uninoculated and inoculated groups of resistant clonal line at four time points.

**Figure 6 genes-13-00788-f006:**
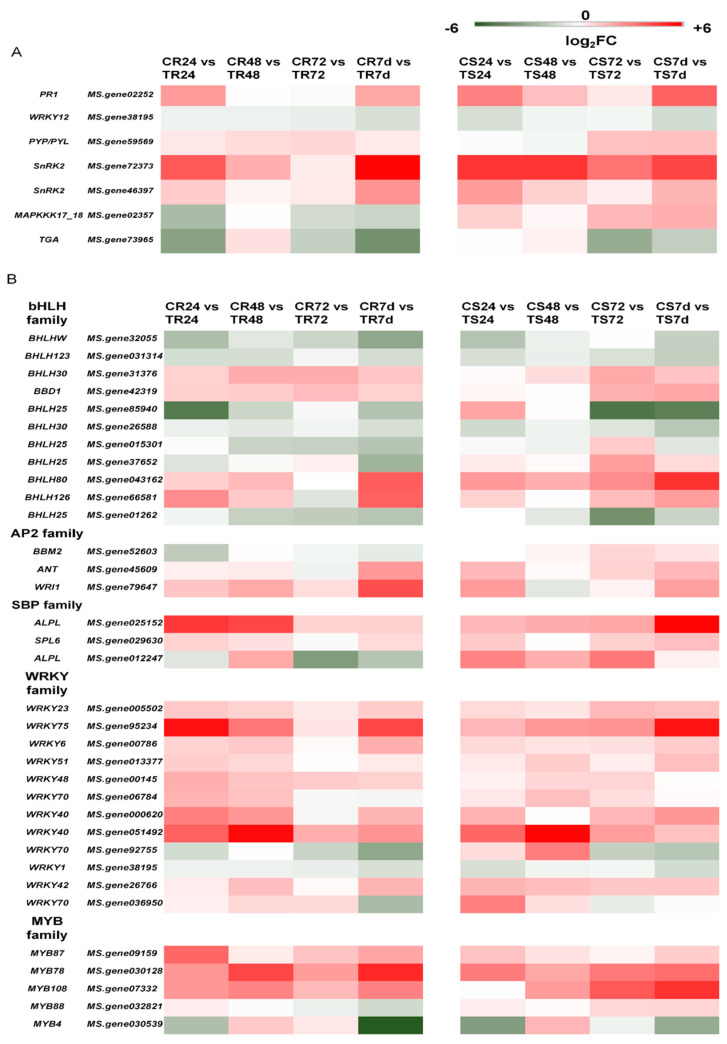
Relative expression of plant–pathogen interaction genes and disease–resistant transcription factor families in two clonal lines. (**A**) Relative expression of plantpathogeninteractionrelated genes in two clonal lines. (**B**) Relative expression of disease–resistant transcription factor families in the two clonal lines.

**Figure 7 genes-13-00788-f007:**
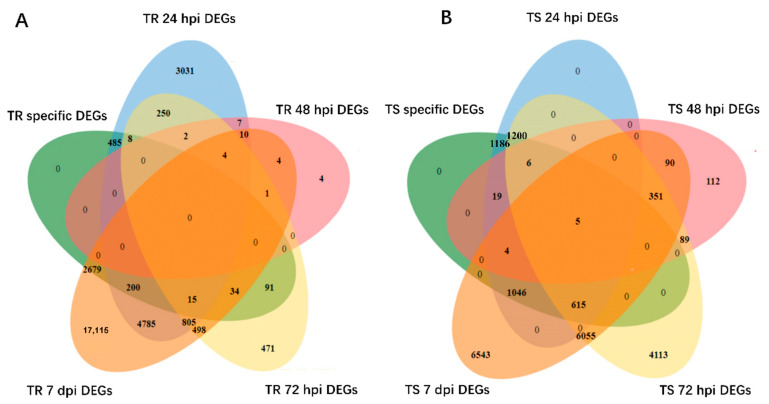
This Venn diagram presents the DEGs of clonal line type-specific transcriptional changes. (**A**) Specific DEGs of resistant clonal line at four time points. (**B**) Specific DEGs of susceptible clonal line at four time points. TR specific DEGs: specific DEGs of resistant clonal line induced by *F. proliferatum* L1 at four time points; TS specific DEGs: specific DEGs of susceptible clonal line induced by *F. proliferatum* L1 at four time points.

**Figure 8 genes-13-00788-f008:**
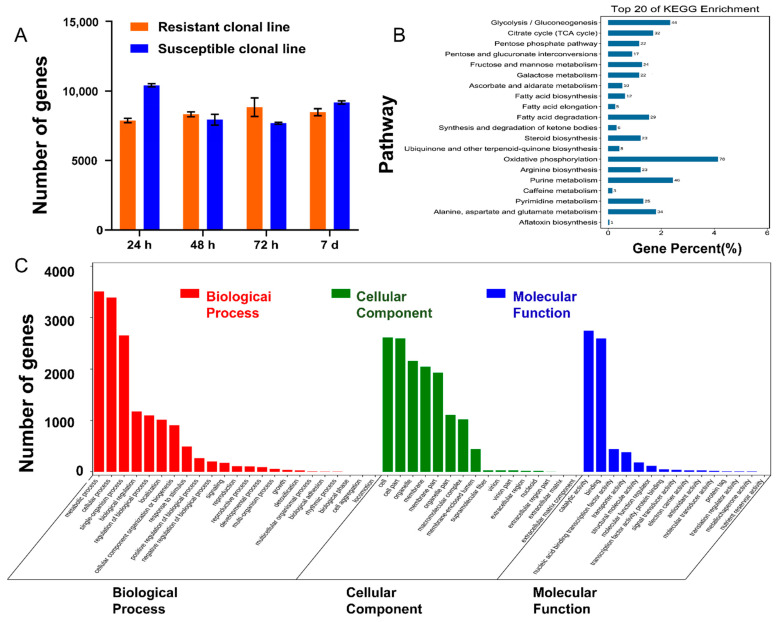
Transcriptome analyses of *F. proliferatum* L1after inoculation. (**A**) Number of fungal genes detected at four time points after inoculation. The values are presented as the mean ± SE. Values of three independent biological replicates per time point. (**B**) KEGG enrichment analyses of genes detected in *F. proliferatum* L1. (**C**) GO enrichment analyses of genes detected in *F. proliferatum* L1. GO, gene ontology; KEGG, Kyoto Encyclopedia of Genes and Genomes; SE, standard error.

**Figure 9 genes-13-00788-f009:**
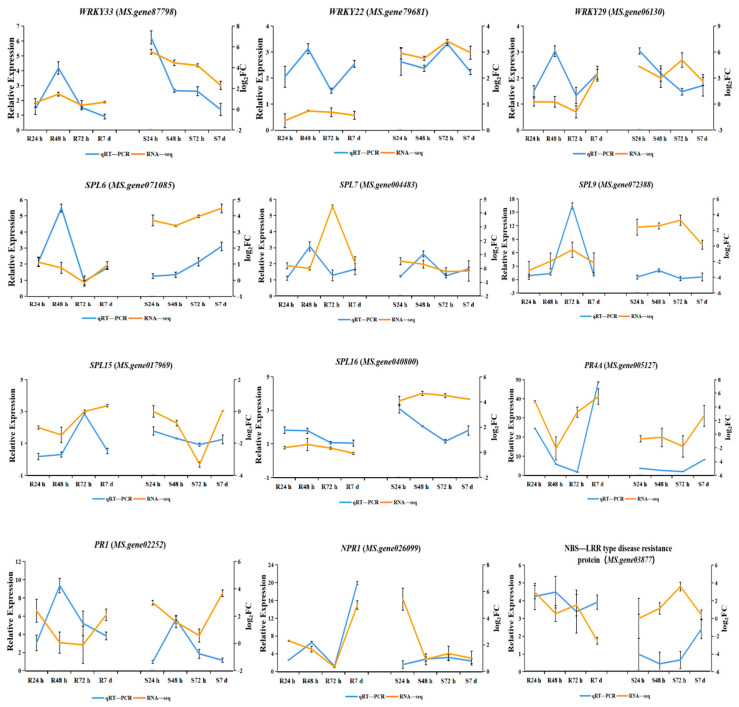
Validation of RNA–seq data by qRT–PCR. Blue lines represent the relative expression levels determined by qRT–PCR (left *y*-axis). Orange lines indicate the log_2_FC based on the read count values of the RNA–seq analysis (right *y*-axis). Error bars indicate standard errors of the means (*n* = 3).

## Data Availability

The original contributions presented for this study can be found in the NCBI database (accession number: PRJNA782344; https://www.ncbi.nlm.nih.gov/bioproject/?term=PRJNA782344 (accessed on 1 April 2022)).

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
