# Peer review of "Transcriptome Analysis of Fusarium Root-Rot-Resistant and -Susceptible Alfalfa (Medicago sativa L.) Plants during Plant–Pathogen Interactions"

_genes, 2022, doi:10.3390/genes13050788_

Round 1

Reviewer 1 Report

The paper describes an extensive study of the cause of the root rot of Alfalfa by Fusaruim spp, specifically F. proliferatum. The strength of this paper is in the wide range of conditions tested to reach satisfactory and wide conclusions: four times points; inoculated vs. uninoculated; resistant vs. susceptible. Implications from the results produced from this paper could be very beneficial for the agriculture field after some subsequent studies and validations. My opinion is that this study is important and relevant.

I have some comments and questions that I would like to get clarification for.

Minor Comments:

Lines 100-103 should be deleted, leftover from paper preparation instructions.

Figure 1, left section, wrong labeling of the Resistant and Susceptible plants.

Lines 117-119, sentence needs to be rewritten, it is not very clear.

Figure 2E, please add statistics to show which expression is significant.

Figures 3, 4 and 5, explain all the acronyms (CS, TR, CK).

The paragraph lines 341-346 needs to be rewritten, it is not clear at all and a little confusing. Is it the number of genes that is changed or the level of expression?

For sections 3.5 and 3.6, I suggest you make pie charts for the percentage of genes. This will make it easier to read and compare the results.

I suggest you move Tables 1 and 2 to Supplementary data.

Lines 395-397, need to be revised and rewritten. The sentence is not clear.

Questions that need some clarifications:

Line 129, why was a hydroponic system used? Could you please briefly describe the reason behind using this system and not soil based system?

Materials and Methods 2.2 section and Results 3.1 sections are very repetitive and should be combined in my opinion. 3.1 is described much more clear than 2.2.

What was the reason behind planting the resistant and susceptible lines together when inoculating? Why not separately?

From which part of the plant was RNA extracted for sequencing? The Entire plant or just the roots? This could affect the results since the initial interaction is in the roots and early time points the gene expression could be masked if RNA was extracted from the whole plant.

Figure 4, why is the number of DEGs expressed so low at the 48 hr and 72 hr time points? Could you please give some explanation as to why that is the case?

Lines 355-361, please provide some background about disease resistant transcription factors, this claim is not really clear since no references were provided.

The entire paper needs proofreading, but especially the Discussion, there are some grammatical errors and unclear sentences.

Author Response

Point 1: Lines 100-103 should be deleted, leftover from paper preparation instructions.

Response 1:  We have deleted  Lines 100-103 in the revised version.

Point 2: Figure 1, left section, wrong labeling of the Resistant and Susceptible plants.

Response 2:  In the revised version, We have modified the Figure 1.

Point 3: Lines 117-119, sentence needs to be rewritten, it is not very clear.

Response 3: We have rewrite this part in revised version.

Point 4: Figure 2E, please add statistics to show which expression is significant.

Response 4: For Figure 2E, we hope to select the sampling time by  marker gene expression, So we focus on the trend of gene expression.

Point 5:Figures 3, 4 and 5, explain all the acronyms (CS, TR, CK).

Response 5:  We add the the acronym meaning in reviesd version.

Point 6: The paragraph lines 341-346 needs to be rewritten, it is not clear at all and a little confusing. Is it the number of genes that is changed or the level of expression?

Response 6: We have rewrite this part. The number of genes include changed and the level of expression.

Point 7: For sections 3.5 and 3.6, I suggest you make pie charts for the percentage of genes. This will make it easier to read and compare the results.

Response7: We  draw the pie chart, but due to many figures in the manuscrpt, so we put it in the appended figure in the revised version.

Point 8: I suggest you move Tables 1 and 2 to Supplementary data.

Response 8:  We move the Table 2 to  Supplementary data. Table 1 shows the information of top DEGs, may give a reference for the researchers some, so put in the text will be better ?

Point 9: Lines 395-397, need to be revised and rewritten. The sentence is not clear.

Response 9: We have rewrite this part.

Point 10: Line 129, why was a hydroponic system used? Could you please briefly describe the reason behind using this system and not soil based system?

Response 10:  There are two main reasons, 1) The consistency of soil culture and soil culture inoculation was poor for transcriptome sequencing; 2) Our sequencing samples is root, hydroponic system would be more convenient and reduce reduce interference.

Point 11: Materials and Methods 2.2 section and Results 3.1 sections are very repetitive and should be combined in my opinion. 3.1 is described much more clear than 2.2.

Response 11:   In this paper, Resistant and susceptible plant were selected as transcriptome materials , this is different from previous transcriptome studies of alfalfa, it has certain particularity and innovation. In order to give some valuable reference to later relevant researchers, So we described the detailed for method in 2.2 section.

Point 12: What was the reason behind planting the resistant and susceptible lines together when inoculating? Why not separately?

Response 12: We inoculate the resistant and susceptible lines separately when sampling for transcriptome sequencing.  But in the second round of hydroponic screening stage, planting the resistant and susceptible lines together in order to enhance the consistency of the screening.

Point 13: From which part of the plant was RNA extracted for sequencing? The Entire plant or just the roots? This could affect the results since the initial interaction is in the roots and early time points the gene expression could be masked if RNA was extracted from the whole plant.

Response 13:  RNA extracted for sequencing was just the roots.

Point 14: Figure 4, why is the number of DEGs expressed so low at the 48 hr and 72 hr time points? Could you please give some explanation as to why that is the case?

Response 14: A large number of primary transcription factors and related genes associated with plant immunity changed 24 h after inoculation in response to Fusarium inoculation, and a large number of secondary metabolism-related genes and downstream signaling-related genes changed 7 days after inoculation due to visible yellowing and defoliation of the plant phenotype. At 48 hpi and 72hpi, the plant response to Fusarium is no longer particularly intense and secondary metabolic pathways have not yet changed significantly, so no significant number of DEGs were detected at these two time points.

Point 15: Lines 355-361, please provide some background about disease resistant transcription factors, this claim is not really clear since no references were provided.

Response 15: We add these information in discussion part in the revised version.

Basic Helix-Loop-Helix (bHLH) transcription factors: The basic helix-loop-helix (bHLH) transcription factor family is one of the largest transcription factor gene families in Arabidopsis thaliana, these transcription factors have the pleiotropic regulatory roles of in plant growth and development, stress response, biochemical functions and the web of signaling networks. ( Hao Y, Zong X, Ren P, et al. Basic Helix-Loop-Helix (bHLH) Transcription Factors Regulate a Wide Range of Functions in Arabidopsis[J]. International Journal of Molecular Sciences, 2021, 22(13):7152.)

APETALA2/ethylene-responsive element binding factors (AP2/ERF) transcription factors: The transcription factors of this family serve as important regulators in many biological and physiological processes, such as plant morphogenesis, responsive mechanisms to various stresses, hormone signal transduction, and metabolite regulation. (Feng K, Hou X L, Xing G M, et al. Advances in AP2/ERF super-family transcription factors in plant[J]. Critical reviews in biotechnology, 2020, 40(6):750-776.)

WRKY transcription factors: WRKY transcription factors are among the largest families of transcriptional regulators. WRKY TFs in adapting plant to a variety of stressed environments. WRKY TFs can regulate diverse biological functions from receptors for pathogen triggered immunity, modulator of chromatin for specific interaction and signal transfer through a complicated network of genes. ( Wani S H, Anand S, Singh B, et al. WRKY transcription factors and plant defense responses: latest discoveries and future prospects[J]. Plant cell reports, 2021, 40(7):1071-1085.)

MYB transcription factors: MYB transcription factors, as one of the most widespread transcription factor families in plants, participate in plant development and responses to stresses by combining with MYB cis-elements in promoters of target genes. MYB transcription factors have been extensively studied and have proven to be critical in the biosynthesis of secondary metabolites in plants, including anthocyanins, flavonols, and lignin. (Wang X, Niu Y,  Zheng Y. Multiple Functions of MYB Transcription Factors in Abiotic Stress Responses[J]. Int J Mol Sci, 2021, 22(11).)

Point 16: The entire paper needs proofreading, but especially the Discussion, there are some grammatical errors and unclear sentences.

Response 16: Thank you for your suggestions, we revised the entire paper  carefully.

Dear reviewer:

On behalf of my co-authors, we are very appreciating reviewer for the positive and constructive comments and suggestions on our manuscript. These suggestions are very important to improve the quality of our manuscripts. We have carefully studied the comments and revised the manuscript, and point-to-point responses to the comments and suggestions of the reviewers. In addition to the reviewer's revision comments, we also revised the manuscript throughout in the revised version.

Sincerely,

Wenyv Zhang (First author)

Lili Cong (Correspondence author)

Reviewer 2 Report

This study uses RNAseq to analyze genes related to Fusarium root rot in alfalfa. The data generated from the resistant and susceptible lines at multiple time points after inoculation is useful for the search for candidate genes for disease resistance. However, the data analysis is not comprehensive and some of the analysis methods are inadequate and outdated. There is also a lack of in-depth analysis of the results. Additionally, this article is poorly written with many grammatical and formatting errors. The experiment design is not clearly described, which makes the article more difficult to read and understand. 

My comments are:

Page 1 line 14: China is a part of the world

Page 1 line 20: Not clear, what does this number mean?

Page 1 line 21: More DEG compared to what? Please be clear.

Page 1 line 31: It’s referred as “queen of forage” in English: Chaffin, W. (2022). Alfalfa: Queen of forage crops. Oklahoma Cooperative Extension Service.

Page 1 line 41: This sentence is confusing.

Page 1 line 42-45: Hard to read, please rewrite. Also need to add the citations of the disease impacts.

Page 2 line 51: It causes increased disease severity rather than “reduced disease resistance in alfalfa”, since the plant immunity is genetically controlled.

Page 2 line 57-59: This sentence is not complete

Page 2 line 71: Hard to read, and please add citation.

Page 2 line 79: Not clear, be specific

Page 2 line 80: “transcriptomic study” -> “transcriptomic studies”

Page 2 line 82: Its confusing, please rewrite

Page 2 line 89: is “a” highly heterozygous autotetraploid plant. Also needs citations here.

Page 3 line 100-103: Delete this part.

Page 3 line 117-118: Please rephrase this sentence.

Page 3 line 128: Be specific about the screening method. What measurements or ratings are used to screen the resistant line?

Page 4 line 134: Selected for what? How many plants are selected?

Page 4 line 136: The title of this section is not accurate. It’s majorly about the qRT-PCR

Page 4 line 138: No spatial data is analyzed in this study.

Page 4 line 141-143: Rewrite. Please be clear and specific about the method and describe the reasoning of gene selection.

Page 4 line 143-147: What is the control group? Is the whole root collected for the experiment? Need to briefly describe the methods of RNA extraction, cDNA synthesis and library construction.

Page 4 line 149: Hard to figure out what are the 48 samples, since the experiment design for RNAseq is not clear.

Page 4 line 148: For the RNAseq data analysis, you need to provide the parameters for reads trimming, alignment, etc. Besides, the proper citations of each software should be added.

Page 4 line 157-159: This part is confusing.

Page 4 line 160: Wrong, FPKM/RPKM is not a suitable measurement for comparing gene between samples. Please see "Misuse of RPKM or TPM normalization when comparing across samples and sequencing protocols." Rna 26.8 (2020): 903-909. It’s better to use TMM (EdgeR) or “Relative Log Expression” normalization (DESeq2) for the analysis.

Page 4 line 164: DESeq is outdated. It’s better to use state-of-art software such as DESeq2 or EdgeR for accurate analysis.

Page 4 line 167: Is the comparison done for each time point of both lines? Please be clear about the RNAseq design.

Page 4 line 170: Annotated by what methods using these databases?

Page 4 line 180-181: Be specific how it is analyzed.

Page 5 line 185: Be specific what samples (time points/genotypes) are used for the qPCR

Page 5 line 189: What’s the model of spectrophotometer? Is cDNA purified? It may be inaccurate to quantify cDNA using spectrophotometer.

Page 5 line 209 and Figure 2: Is 0h used as control? Looks like NPR3 is not significantly changed at 24hpi. Please add the statistical significance level of expression differences to the figure. It’s better to add the picture showing the root phenotype, especially for the time points used for RNAseq.

Page 5 line 211: Use “hpi” or “dpi”. “12 h” is not a time point.

Page 6 line 224: “8Gb” is too little for 48 samples. Please double check. The raw reads are not reported and the trimming is not described, the percentage of clean reads reported here is meaningless.

Page 6 line 227: What’s the ratio of uniquely mapped reads?

Page 6 line 230: What type of filtering? The analysis on the raw counts here is not useful since the expressed gene composition/sequencing depth difference hasn’t been normalized. Do the analysis such correlation and PCA using the normalized reads.

Page 6 line 232: Be clear if the gene number is the total number identified in 48 samples or average numb. If these they are all expressed genes, what is the threshold to call an expressed gene? This need to be clarified.

Page 6 line 234: The title is confusing. And what are the two lines?

Page 6 line 237: This sentence is too long and confusing. Please rewrite. GO analysis is not used for detecting DEGs, and the accuracy heavily depends on the quality of the transcriptome assembly and annotation.

Page 6 line 238: What do these IDs mean? Explain in the materials and methods.

Page 6 line 246: Be specific about the result. How is the DEGs different. Most importantly, Explain why you compared the GO/KEGG terms of these two groups in this paragraph? You will probably find similar terms in other groups. If not, what’s the difference between other groups? The DEGs identified in the uninoculated groups should be the genes contributed to the genetic difference between the susceptible and resistance lines. The DEGs identified in the inoculated groups should include important R genes and genes contributed to the genetic difference. You may identify key R genes by comparing the DEGs of the two types. It should be thoroughly analyzed here.

Page 6 line 247: Be specific, and avoid using “between two lines”.

Page 6 line 251: This sentence is confusing.

Page 6 line 256: The statement is not convincing if it is only based on the genes overlapped at all time points. What about the genes that showed higher expression at 3 time points but similar at 1 time point?

Page 6 line 258: Based on what?

Page 6 line 259: What are “these DEGs”?

Page 6 line 266: Add “and” before “MS.gene98660.”

Page 7 line 267-270: Split the sentence to two. Besides, where is the result showing stronger cell wall in resistant lines?

Page 7 line 273: Put it to discussion and add citations.

Page 7 line 275: Are these DEGs identified by comparing R and S lines in the uninoculated groups? At what time point are they identified? You later mentioned 1,057 DEGs are enriched. What are those genes? The whole paragraph is confusing. The RNAseq design should be clearly described in the methods and the types of DEGs should be clarified when you present the result.

Page 7 line 284: The DEGs identified in the inoculated groups are very important, but there is no comprehensive analysis.

Page 7 line 298 – Page 8 line 318: Please rephrase. Be consistent about the number formatting, e.g. 4,475 vs 5127.

Page 7 Figure 4: It’s weird that only a few DEGs are detected at 48 hpi while there are thousands of DEGs detected at 24hpi. This raises serious questions about the validity of the data (from sampling to library construction to analysis.). Please look into this result. It would also be helpful to present QC analysis such as PCA and Hierarchical Clustering.

Page 8 line 320: Do the functional analysis for each time point because they belong to different pathogenesis stages (colonization, infection, etc.)

Page 8 line 327: How are they identified and removed? Be cautious about this step because you may delete important candidate genes.

Page 8 line 334-336: This sentence is confusing.

Page 9 Figure 5: The abbreviations are not explained. And the description is confusing.

Page 9 line 341: How is the number of DEGs increased or decreased?

Page 9 line 348: How to define “co-modulated”?

Page 9 line 348-349: How many genes are annotated by KEGG? The KEGG annotation is sometimes not complete, therefore, generating inaccurate reference. Since one of the goals of this study is to identify novel genes. Why excluding the genes with unspecified functions? There are plenty bioinformatic methods (e.g., co-expression and phylogenetic analysis) to predict the functions of these genes.

Page 10 line 369: For this type of analysis, I recommend using the DESeq2 Likelihood Ratio Test to do the time course analysis, which will find the genes reacted differently to the inoculation over time. Please see: https://master.bioconductor.org/packages/release/workflows/vignettes/rnaseqGene/inst/doc/rnaseqGene.html#time-course-experiments

Page 11 line 381: What’s the difference? Is it only the gene number? What about the difference in gene regulation direction (up or down regulated)?

Page 11 line 383: How is the top DEGs selected? How to justify the analysis on the two lines by only using only these top 30 DEGs. Besides, fewer DEGs identified at other time points doesn’t mean that they are not important. There are still large gene number difference at 72hpi.

Page 11 line 411: For this analysis, it is necessary to report the number of reads aligned to the pathogen genome, the coverage of the transcriptome. The proper normalization is also very important since the fungi abundance in the plant will affect the quantification.

Page 12 line 440: Re-analyze RNAseq using other units.

Page 18 line 488: Need to revise the discussions based on the updated results.

Page 18 line 492: There is no spatial data analyzed.

Page 18 line 495-496: This sentence is confusing, rewrite.

Page 18 line 521: To better illustrate the plant immunity response, you can map the DEGs to the KEGG Plant-pathogen interaction.

Page 18 line 534: The statement is not convincing based on Table S7.  Besides, Table S7 need to be improved

Page 19 line 557: Be specific about the expressed genes. For example, how many genes are significant DEGs (either up or downregulated) in resistant lines. It’s better to use fold change to interpret the result than comparing the FPKM value.

Page 19 line 586: Where is the result?

Page 20 line 626: Molecular mechanisms are not identified in this study.

Author Response

Point 1: Page 1 line 14: China is a part of the world

Response 1: We  revised this sentence in the new version.

Point 2: Page 1 line 20: Not clear, what does this number mean?

Response 2: . Among all the differentially expressed genes (DEGs) detected between the two clonal lines at the four time points after inoculation, approximately 81.8% of DEGs  were detected at 24h and  7d after inoculation.

Point 3: Page 1 line 21: More DEG compared to what? Please be clear.

Response 3: DEG refers to the two inoculated clonal lines, revised in new version.

Point 4: Page 1 line 31: It’s referred as “queen of forage” in English: Chaffin, W. (2022). Alfalfa: Queen of forage crops. Oklahoma Cooperative Extension Service.

Response 4: We revised to  “queen of forage”.

Point 5: Page 1 line 41: This sentence is confusing.

Response 5: We rewrite the sentence in the revised version.

Point 6: Page 1 line 42-45: Hard to read, please rewrite. Also need to add the citations of the disease impacts.

Response 6. We rewrite the sentence and add the citations in the revised version.

Point 7: Page 2 line 51: It causes increased disease severity rather than “reduced disease resistance in alfalfa”, since the plant immunity is genetically controlled.

Response 7: We rewrite the sentence in the revised version

Point 8: Page 2 line 57-59: This sentence is not complete

Response 8: We rewrite the sentence in the revised version.

Point 9: Page 2 line 71: Hard to read, and please add citation.

Response 9: We rewrite the sentence in the revised version.

Point 10: Page 2 line 79: Not clear, be specific

Response 10: We rewrite the sentence in the revised version.

Point 11: Page 2 line 80: “transcriptomic study” -> “transcriptomic studies”

Response11: We changed to “transcriptomic studies” in reviesd version.

Point 12: Page 2 line 82: Its confusing, please rewrite

Response 12: We rewrite the sentence in the revised version.

Point 13: Page 2 line 89: is “a” highly heterozygous autotetraploid plant. Also needs citations here.

Response 13: For this sentence, reference [33] included the content” highly heterozygous autotetraploid plant”, so we juast citaion in the end of the sentence.

Point 14: Page 3 line 100-103: Delete this part.

Response 14: We delete this part in the revised version.

Point 15: Page 3 line 117-118: Please rephrase this sentence.

Response 15: We rewrite the sentence in the revised version.

Point 16: Page 3 line 128: Be specific about the screening method. What measurements or ratings are used to screen the resistant line?

Response 16:Screen by root disease spot size and phenotype at 45 d after inoculation

Point 17: Page 4 line 134: Selected for what? How many plants are selected?

Response 17: We rewrite the sentence in the revised version. About two rounds of screening results we put in the 3.1.

Point 18: Page 4 line 136: The title of this section is not accurate. It’s majorly about the qRT-PCR

Response 18: We modifeid the title in the revised version.

Point 19: Page 4 line 138: No spatial data is analyzed in this study.

Response 19: We rewrite the sentence in the revised version

Point 20: Page 4 line 141-143: Rewrite. Please be clear and specific about the method and describe the reasoning of gene selection.

Response 20: NPR1 and NPR3 genes in Arabidopsis thaliana has been identified as important disease resistance genes. SPL15 gene in alfalfa is the IPA1 homologous gene of rice and has been  reported as  disease resistance genes. In our previous study, we also verify the SPL15 to participate in the process of alfalfa root rot resistance (Data has not been published).

Point 21: Page 4 line 143-147: What is the control group? Is the whole root collected for the experiment? Need to briefly describe the methods of RNA extraction, cDNA synthesis and library construction.

Response 21: Uninoculated group as control group,   add briefly describe the methods of RNA extraction, cDNA synthesis and library construction in the revised version.

Point 22: Page 4 line 149: Hard to figure out what are the 48 samples, since the experiment design for RNAseq is not clear.

Response 22:  12 samples of the resistant clonal lines uninoculated group at four time points, 12 sam-ples of the susceptible clonal lines inoculated group at four time points and 12 samples of the susceptible clonal lines uninoculated group at four time points for a total of 48 samples

Point 23: Page 4 line 148: For the RNAseq data analysis, you need to provide the parameters for reads trimming, alignment, etc. Besides, the proper citations of each software should be added.

Response 23: We rewrite this part and add the citation in the revised version.

Point 24: Page 4 line 157-159: This part is confusing.

Response 24: We rewrite the sentence in the revised version

Point 25:  Page 4 line 160: Wrong, FPKM/RPKM is not a suitable measurement for comparing gene between samples. Please see "Misuse of RPKM or TPM normalization when comparing across samples and sequencing protocols." Rna 26.8 (2020): 903-909. It’s better to use TMM (EdgeR) or “Relative Log Expression” normalization (DESeq2) for the analysis.

Response 25: According to the opinion of the sequencing company and  the reference (Transcriptome analysis reveals the promotive effect of potassium by hormones and sugar signaling pathways during adventitious roots formation in the apple rootstock ), To find differentially expressed genes (DEGs), according to fragments per kilobase of exon per million mapped reads (FRKM). We used DESeq2 normalization for the analysis.  

Point 26: Page 4 line 164: DESeq is outdated. It’s better to use state-of-art software such as DESeq2 or EdgeR for accurate analysis.

Response 26: This is a spelling mistake, we used DESeq2.

Point 27: Page 4 line 167: Is the comparison done for each time point of both lines? Please be clear about the RNAseq design.

Response 27: Comparison done for each time point of both lines.

Point 28: Page 4 line 170: Annotated by what methods using these databases?

Response 28: We rewrite this part.

Point 29: Page 4 line 180-181: Be specific how it is analyzed.

Response 29:  Because the fungus may invade the root or adhere to the root surface, so there are both alfalfa and fungus sequence in the transcriptome data.

Point 30: Page 5 line 185: Be specific what samples (time points/genotypes) are used for the qPCR

Response 30: We rewrite this part. Total RNA of 48 samples subjected to transcriptome sequencing for qRT-PCR

Point 31: Page 5 line 189: What’s the model of spectrophotometer? Is cDNA purified? It may be inaccurate to quantify cDNA using spectrophotometer.

Response 31: NanoDrop One (Thermo Fisher Scientific) were used for quantitative purified cDNA.

Point 32: Page 5 line 209 and Figure 2: Is 0h used as control? Looks like NPR3 is not significantly changed at 24hpi. Please add the statistical significance level of expression differences to the figure. It’s better to add the picture showing the root phenotype, especially for the time points used for RNAseq.

Response 32:  we add the statistical significance level. Because with the three genes, so we focus on the  trends, it is difficult to meet the three genes have significant change at the same time. For the picture, Because disease phenotype after inoculation is not as rapid and significant as salt stress, the root of the disease spot is lesser, photo effect is not good, so we didn't photo the roots.

Point 33: Page 5 line 211: Use “hpi” or “dpi”. “12 h” is not a time point.

Response 33: we change to “hpi” or “dpi” in the revised version.

Point 34: Page 6 line 224: “8Gb” is too little for 48 samples. Please double check. The raw reads are not reported and the trimming is not described, the percentage of clean reads reported here is meaningless.

Response 34: Each sample for “8Gb”,

Point 35: Page 6 line 227: What’s the ratio of uniquely mapped reads?

Response 35:  Because of Xinjiangdaye sequenced completetly for sister chromatids, and  we have high sequencing depth (8Gb for each sample), so the ratio of uniquely mapped reads is 52.2%.

Point 36: Page 6 line 230: What type of filtering? The analysis on the raw counts here is not useful since the expressed gene composition/sequencing depth difference hasn’t been normalized. Do the analysis such correlation and PCA using the normalized reads.

Response 36: . For accurate analysis, pre-processing of RNA sequence raw data was performed using the Cutadapt (https://cutadapt.readthedocs.io/en/stable/)to removal of 3 'end adapter,  and removing reads with average quality below Q20 and minimum read size (50 bp). correlation and PCA using the normalized reads.

Point 37: Page 6 line 232: Be clear if the gene number is the total number identified in 48 samples or average numb. If these they are all expressed genes, what is the threshold to call an expressed gene? This need to be clarified.

Response 37: gene number is the total number identified in 48 samples. the screening conditions were p-adjusted < 0.05 and log2|FC|≥1

Point 38: Page 6 line 234: The title is confusing. And what are the two lines?

Response 38: Resistant and susceptible  two clonal lines.

Point 39: Page 6 line 237: This sentence is too long and confusing. Please rewrite. GO analysis is not used for detecting DEGs, and the accuracy heavily depends on the quality of the transcriptome assembly and annotation.

Response 39: We rewrite this part.

Point 40: Page 6 line 238: What do these IDs mean? Explain in the materials and methods.

Response 40: We explain the meaning in Fig. Caption in revised version. CS: uninoculated groups of susceptible clonal line; TS: inoculated groups of susceptible clonal line; CR: uninoculated groups of resistant clonal line; TR: inoculated groups of resistant clonal line.

Point 41: Page 6 line 246: Be specific about the result. How is the DEGs different. Most importantly, Explain why you compared the GO/KEGG terms of these two groups in this paragraph? You will probably find similar terms in other groups. If not, what’s the difference between other groups? The DEGs identified in the uninoculated groups should be the genes contributed to the genetic difference between the susceptible and resistance lines. The DEGs identified in the inoculated groups should include important R genes and genes contributed to the genetic difference. You may identify key R genes by comparing the DEGs of the two types. It should be thoroughly analyzed here.

Response 41: Because the data is too large and complex, so we chose the two points (24hpi and 48hpi ) with more  DEGs were analyzed.

Point 42: Page 6 line 247: Be specific, and avoid using “between two lines”.

Response 42:  We rewrite this sentence.

Point 43: Page 6 line 251: This sentence is confusing.

Response 43: We rewrite this sentence.

Point 44: Page 6 line 256: The statement is not convincing if it is only based on the genes overlapped at all time points. What about the genes that showed higher expression at 3 time points but similar at 1 time point?

Response 44: We rewrite this part.

Point 45: Page 6 line 258: Based on what?

Response 45: Based on DEGs analysis before inoculation (CS vs CR)

Point 46: Page 6 line 259: What are “these DEGs”?

Response 46: DEGs of two uninoculated clonal lines (CS vs CR)

Point 47: Page 6 line 266: Add “and” before “MS.gene98660.”

Response 47:  added in revised version.

Point 48: Page 7 line 267-270: Split the sentence to two. Besides, where is the result showing stronger cell wall in resistant lines?

Response 48: We rewrite this part.

Point 49: Page 7 line 273: Put it to discussion and add citations.

Response 49: We rewrite this part.

Point 50: Page 7 line 275: Are these DEGs identified by comparing R and S lines in the uninoculated groups? At what time point are they identified? You later mentioned 1,057 DEGs are enriched. What are those genes? The whole paragraph is confusing. The RNAseq design should be clearly described in the methods and the types of DEGs should be clarified when you present the result.

Response 50: We rewrite this part. these DEGs identified by comparing R and S lines in the uninoculated groups

Point 51: Page 7 line 284: The DEGs identified in the inoculated groups are very important, but there is no comprehensive analysis.

Response 51: We put the DEGs identified in the inoculated groups in the 3.4.

Point 52: Page 7 line 298 – Page 8 line 318: Please rephrase. Be consistent about the number formatting, e.g. 4,475 vs 5127.

Response 52: We revised this part.

Point 53: Page 7 Figure 4: It’s weird that only a few DEGs are detected at 48 hpi while there are thousands of DEGs detected at 24hpi. This raises serious questions about the validity of the data (from sampling to library construction to analysis.). Please look into this result. It would also be helpful to present QC analysis such as PCA and Hierarchical Clustering.

Response 53: We also noticed this phenomenon, and we check all the steps including the extraction of RNA, and correlation analysis and DEGs analysis, and no errors were reported in quality control data and analysis process.We found a few DEGs are detected at 48 hpi both in (CRvs TR) and (CS vs TS), the results are consistent at this time point. We hope can be validated this special phenomenon in the later research and clarify the reason.

Point 54: Page 8 line 320: Do the functional analysis for each time point because they belong to different pathogenesis stages (colonization, infection, etc.)

Response 54: The functional analysis for each time point were conducted, but emphatically analyzed samples at 24hpi and 7 dpi.

Point 55: Page 8 line 327: How are they identified and removed? Be cautious about this step because you may delete important candidate genes.

Response 55: Using FunRich software ( http://www.funrich.org/) remove the genetically different genes. These genes from genetic differences are also our concerned genes,But it is hard to analysis with inoculation group. So we analyzed them separately. In 3.2, we analyzed DEGs between uninoculated susceptible clonal line  and resistant clonal line. 

Point 56: Page 8 line 334-336: This sentence is confusing.

Response 56: We deleted this sentence.

Point 57: Page 9 Figure 5: The abbreviations are not explained. And the description is confusing.

Response 57: we add the abbreviations and rewrite description in the revised version.

Point 58: Page 9 line 341: How is the number of DEGs increased or decreased?

Response 58: It is wrong description,  We revised this sentence.

Point 59: Page 9 line 348: How to define “co-modulated”?

Response 59: We revised this part.

Point 60: Page 9 line 348-349: How many genes are annotated by KEGG? The KEGG annotation is sometimes not complete, therefore, generating inaccurate reference. Since one of the goals of this study is to identify novel genes. Why excluding the genes with unspecified functions? There are plenty bioinformatic methods (e.g., co-expression and phylogenetic analysis) to predict the functions of these genes.

Response 60: about 500 genes are annotated by KEGG.

Point 61: Page 10 line 369: For this type of analysis, I recommend using the DESeq2 Likelihood Ratio Test to do the time course analysis, which will find the genes reacted differently to the inoculation over time. Please see: https://master.bioconductor.org/packages/release/workflows/vignettes/rnaseqGene/inst/doc/rnaseqGene.html#time-course-experiments

Response 61: We used DESeq2 software.

Point 62: Page 11 line 381: What’s the difference? Is it only the gene number? What about the difference in gene regulati Is it only the gene number on direction (up or down regulated)?

Response 62: the difference is DEGs number. Because the data in this manuscrpit  is too large and complex, we hardly to explain more details in this article. So we try to analyze some obvious results in here. About the difference in gene regulation, we show the top 30 genes in differential expression levels of the two clonal lines at 24h and 7d (Table 1).

Point 63: Page 11 line 383: How is the top DEGs selected? How to justify the analysis on the two lines by only using only these top 30 DEGs. Besides, fewer DEGs identified at other time points doesn’t mean that they are not important. There are still large gene number difference at 72hpi.

Response 63: Based on Foldchange. We are making the difference  analysis for all time points, but it is hard to list all for us.

Point 64: Page 11 line 411: For this analysis, it is necessary to report the number of reads aligned to the pathogen genome, the coverage of the transcriptome. The proper normalization is also very important since the fungi abundance in the plant will affect the quantification.

Response 64: We rewrite this part.

Point 65: Page 12 line 440: Re-analyze RNAseq using other units.

Response 65: We rewrite this part.

Point 66: Page 18 line 488: Need to revise the discussions based on the updated results.

Response 66: We rewrite this part.

Point 67: Page 18 line 492: There is no spatial data analyzed.

Response 67: delete  spatial.

Point 68: Page 18 line 495-496: This sentence is confusing, rewrite.

Response 68: We rewrite this sentense.

Point 69: Page 18 line 521: To better illustrate the plant immunity response, you can map the DEGs to the KEGG Plant-pathogen interaction.

Response 69: We map the DEGs to the KEGG Plant-pathogen interactio (https://www.kegg.jp/pathway/map04626)

Point 70: Page 18 line 534: The statement is not convincing based on Table S7.  Besides, Table S7 need to be improved

Response 70: Table S7 improved in revised version.

Point 71: Page 19 line 557: Be specific about the expressed genes. For example, how many genes are significant DEGs (either up or downregulated) in resistant lines. It’s better to use fold change to interpret the result than comparing the FPKM value.

Response 71: We rewrite this part.

Point 72: Page 19 line 586: Where is the result?

Response 72: We delete this  part.

Point 73: Page 20 line 626: Molecular mechanisms are not identified in this study.

Response 73: We revised this part.

Dear reviewer:

On behalf of my co-authors, we are very appreciating reviewer for the positive and constructive comments and suggestions on our manuscript. These suggestions are very important to improve the quality of our manuscripts. We have carefully studied the comments and revised the manuscript, and point-to-point responses to the comments and suggestions of the reviewers. In addition to the reviewer's revision comments, we also revised the manuscript throughout in the revised version.

Sincerely,

Wenyv Zhang (First author)

Lili Cong (Correspondence author)

Round 2

Reviewer 2 Report

Some of my comments are addressed in the updated manuscript, however, the analysis is still not improved and many of my concerns are not resolved. The presentation of the result is not complete, and it will cause confusions for the readers. Besides, there are still many grammar and formatting errors in the manuscript. While some of my previous comments still apply, here are my new comments based on the updated manuscript: 

Page 2 line 111: “slowly” -> “slow”.

Page 3 line 301: Still not specific, how do you define “resistant” or “susceptible” by these measurement.

Page 3 line 307: Delete “the”.

Page 4 line 330: Why the trends of gene expression? And what genes is required? Better to use expression like: In order to …, we examined the expression of defense related genes at different time point after inoculation using qPCR.

Page 4 line 341: Should be according to the result of qPCR, or according to the gene expression level. Incomplete sentence, who determines the time?

Page 4 line 348: Need to rewrite this sentence. Besides, the 48 samples are explained in the authors’ response, but not in the article.

Page 4 line 358: It’s better to cite the publications instead of websites.

Page 4 line 363: The link is broken.

Page 4 line 364: As I mentioned, FPKM shouldn’t be used. Multiple studies have reported the pitfall of FPKM. The author who introduced FPKM also explained why FPKM/RPKM is not good for RNAseq (https://www.youtube.com/watch?v=5NiFibnbE8o#t=1831). And the DESeq2 only requires raw reads as an input.

Page 4 line 370: What are the “basic DEGs”?

Page 4 line 371: Missing spacing in “linesresponding”.

Page 5 line 545: What do you mean by saying “conducted between DEG pairs”? The analysis should be done for a set of DEGs (identified from different comparison groups/pairs).

Page 5 line 550: Hard to read, rewrite.

Page 5 line 551-552: This sentence is incomplete and confusing.

Page 5 line 557: Broken sentence or typo.

Page 5 line 579: Still, NPR3 is not significantly changed. Need to rephrase.

Page 6 Figure 2: While the authors claimed they added statistical level (of multiple comparison test), it is still not included in Figure 2E.

Page 6 line 644: The authors still have not addressed my concern. A normalized reads count should be used for the analysis. And please do the sample-to-sample distances analysis or principal component analysis (PCA) to show the samples/replicates distances. Please refer to DESeq2 vignette for the data quality assessment analysis: https://bioconductor.org/packages/devel/bioc/vignettes/DESeq2/inst/doc/DESeq2.html#data-quality-assessment-by-sample-clustering-and-visualization.

Page 6 line 647: Missing space in “groupsof”.

Page 6 line 650: Need to explain the ID in methods, not just in the Figure of results.

Page 6 line 662: Confusing, and how come there are 592,748,741 and 607 DEGs in 1057 DEGs?

Page 7 line 696: DEGs of all time points? Need to be very clear when describing the DEGs, since you have many types of DEGs in the result.

Page 7 line 709: How. Higher in resistant line?

Page 7 line 710: Remove “were found to be”. Do the 27 DEGs belong to 1057 DEGs? Did you do the enrichment for this small set (27) of DEGs? If the enrichment analysis is done on all DEGs, and it is only the 27 DEGs are enriched in these terms, you should describe the 1,057 DEGs GO enrichment first, then zoom in to focus on small group of DEGs that showed interesting functions.

Page 7 line 722-729: The authors responded by saying that DEGs identified in the inoculated groups were in the 3.4, but these DEGs, identified by comparing resistant and susceptible lines after inoculation, are not analyzed there.

There are three ways of analyzing this type of data. 1) compare the R vs S lines after inoculation at each time point, then filter those in the uninoculated group (genotype difference) to find DEGs related to inoculation.

2) compare the inoculated vs uninoculated at each time point for both R and S lines.  Then compare the pathogen responding DEGs between R and S.

3) use DESeq2 Likelihood Ratio Test to do the time course analysis, which will find the genes reacted differently to the inoculation over time.

They analyzed the strategy 1) in 3.3, but the analysis is incomplete. The 2) method is described in 3.4. I also suggested method 3), but the authors ignored.

Page 9 line 779: You still need to report or mention the functional analysis for each time point. You cannot just report results that are easy for you to analyze or present. Need to add the tables of GO and KEGG enrichment results.

Page 9 line 785: Are they the DEGs of the uninoculated group? Are they the DEGs identified at all time points? Be specific. Funrich is a software for functional enrichment and interaction network analysis. Is it just used to remove overlapped DEGs? Besides, the process should be mentioned in the materials and methods part.

Pape 9 line 789: Figure 5 doesn’t explain this statement. It is shown in Figure 4 A,C.

Page 9 line 790: Not convincing, the recognition of the pathogen or the immunity gene response shouldn’t be delayed. They should carefully analyze the DEGs or add microscopic analysis in order to draw such conclusion.

Page 10 line 833: I would say the TS (4081) and TR (3512) specific DEGs are more important, especially the 3512 DEGs. They are the reason the R lines show higher resistance.

Besides, since the DEGs are analyzed differently, the expression direction of the 2,078 overlapped DEGs (up or down regulated) may be different in the R and S lines. So, I don’t find the enrichment analysis on all overlapped genes useful.

Page 10 line 834: Why no GO enrichment analysis here, The KEGG result is not representative since only 500 out of 2078 DEGs are analyzed.

Page 12 line 870: The difference is still not explained here. It might cause confusions for readers.

Page 12 line 873: The authors didn’t address my concern here. There must be a better way to present and interpret the data. 

Page 12 line 912: The authors didn’t re-analyze the data with correct normalization, as FPKM is still used in the table. They just chose a different word here to hide this issue.

The page index is wrong after Table 1.

In line 1225: They still didn’t address my comment. This study did not identify the molecular mechanisms.
